# Does the Necrotic Portion of Metastatic Lymphadenopathy from Squamous Cell Carcinoma Still Have Tumoral Oncologic Information? Differential Diagnosis of Benign Necrotic Lymphadenopathy Using microRNA

**DOI:** 10.3390/biomedicines11092407

**Published:** 2023-08-28

**Authors:** Eun Shin, Seung Hoon Han, Il-Seok Park, Jee Hye Wee, Joong Seob Lee, Heejin Kim

**Affiliations:** 1Department of Pathology, Dongtan Sacred Heart Hospital, Hwaseong 18450, Republic of Korea; sea4197@hallym.or.kr; 2Department of Otorhinolaryngology-Head and Neck Surgery, Dongtan Sacred Heart Hospital, Hwaseong 18450, Republic of Korea; mip115@hallym.or.kr (S.H.H.); ispark@hallym.or.kr (I.-S.P.); 3Department of Otorhinolaryngology-Head and Neck Surgery, Hallym University Sacred Heart Hospital, Anyang 14068, Republic of Korea; weejh07@hallym.or.kr (J.H.W.); apniosio@hallym.or.kr (J.S.L.)

**Keywords:** intranodal necrosis, metastasis, tuberculosis, microRNA, core needle biopsy

## Abstract

Neck necrotic lymph nodes commonly correspond to metastasis or benign inflammatory conditions such as Kikuchi disease and tuberculosis. Ultrasound-guided biopsy can be used for differential diagnosis, but results may be unclear. Therefore, this study aimed to identify target microRNAs (miRNAs) and genes for the differential diagnosis of inflammatory and malignant necrotic lymph nodes. We selected six inflammatory lymphadenitis formalin-fixed paraffin-embedded (FFPE) samples that showed internal necrosis and five cancer necrotic FFPE samples. Tissue microarray (TMA) was performed to separate the necrotic and cancerous portions. Total RNA was extracted from six pairs of separated inflammatory necrosis, five pairs of cancer necrosis, and cancer portions. Differentially expressed miRNAs were analyzed by comparing inflammatory necrosis, cancer, and cancer necrosis. Seventeen miRNAs were upregulated in cancer necrosis compared to inflammatory necrosis, and two miRNAs (hsa-miR-155-5p and hsa-miR-146b-5p) showed lower expression in cancer necrotic cells. Nineteen miRNAs that were differentially expressed between inflammatory and cancer necrosis were analyzed for target gene expression; these transcripts demonstrated a clear relationship with cancer. The differentially expressed miRNAs in inflammatory and tumor necrosis were associated with cancer-related pathways. These preliminary results might help in the differential diagnosis of cervical metastatic necrotic lymphadenopathy and avoiding unnecessary excisional biopsies.

## 1. Introduction

The diagnosis of cervical lymphadenopathy (LAP) is challenging for physicians. Imaging studies, such as computed tomography (CT) or ultrasonography (US), are required to make an exact diagnosis. In particular, when necrosis is observed within the enlarged lymph nodes in imaging studies, pathologic conditions such as inflammatory disease or tumor infiltration must be considered in the differential diagnosis. Necrotic lymph nodes usually correspond to squamous cell carcinoma (SqCC) metastasis or lymphoma. However, these should also be considered as possible differential diagnoses in specific endemic areas of tuberculosis and Kikuchi disease [1,2,3].

Neck US is the most commonly used tool for initial diagnosis. It is also used for imaging purposes as well as for biopsy guidance. Fine-needle aspiration cytology (FNAC) is a relatively non-invasive and useful method for the initial evaluation of cervical LAP because of its high cancer diagnostic accuracy [4,5]. However, FNAC has the limitations of being non-diagnostic and yielding insufficient samples, especially for lymphoma. Core needle biopsy (CNB) is beneficial for obtaining sufficient tissue and enhancing diagnostic accuracy. However, in some cases, such as severe necrosis or calcification, CNB can also show unclear results.

Necrosis is the pattern of cell death that occurs in response to injuries such as hypoxia or infection. Therefore, determining the exact cell morphology in a small biopsy sample can be difficult when severe necrosis progresses. A biopsy should be performed for a differential diagnosis of necrotic lymph nodes. FNAC or CNB is commonly used for the initial biopsy to avoid interference with future treatment plans. However, many cases of small biopsies in necrotic lymph nodes only report necrotic materials and are not useful for differential diagnosis. Furthermore, many cases of severely necrotic lymph nodes have irregular margins and infiltrate adjacent tissues. Therefore, it is difficult to resect them clearly without any injury to other tissues. In addition, incisional biopsy is usually not favored because of tumor spillage. To avoid unnecessary resection of vital tissues, we hypothesized that the necrotic portion of the tumor can still contain information about the tumor, and miRNAs can help make a differential diagnosis.

MicroRNAs (miRNAs) are non-coding single-stranded RNAs that play a significant role in cancer development, progression, and diagnosis. miRNAs have been studied in various cancers, especially thyroid cancer; a diagnostic tool using miRNAs has been developed and commercially used [6,7]. A previous study investigated the feasibility of using miRNA expression to detect metastatic cells in formalin-fixed paraffin-embedded (FFPE) lymph nodes [8] and hypothesized that necrosis from tumor progression might provide information about cancer despite its pathology results only showing necrosis. Therefore, in this study, we aimed to identify target miRNAs and genes for the differential diagnosis of inflammatory and malignant necrotic lymph nodes that cannot be determined with histopathology.

## 2. Materials and Methods

### 2.1. Patients and Materials—Construction of Tissue Microarray

This study was approved by the Institutional Review Board of Dongtan Sacred Heart Hospital, Hallym University (IRB number 2021-09-014-002), and the committee waived the need for written informed consent. We retrospectively reviewed the CT scans of patients diagnosed with lymph node metastasis of SqCC, tuberculosis, and Kikuchi disease between 2014 and 2022. Among them, we selected patients who showed marked necrosis on CT and underwent lymph node excision or dissection. The original hematoxylin-eosin (H&E)-stained slides were reviewed by a head and neck pathologist (E.S.), and FFPE slides that showed focal necrosis to clear dissection were excluded. Five patients with necrotic lymph node metastasis of SqCC and six patients with inflammatory LAP with necrosis were enrolled following a pathological slide review (Figure 1).

### 2.2. Construction of Tissue Microarray (TMA) and RNA Isolation

Total RNA was extracted from paraffin-embedded tissues using TRIzol reagent (Invitrogen, Carlsbad, CA, USA) according to the manufacturer’s instructions. After deparaffinization with xylene, tissue sections were stained with H&E, and cancer necrosis, cancer, and inflammatory necrosis portions were selected and carefully dissected to minimize contamination. When the necrotic portion was focal, necrosis was clearly dissected using representative core tissue sections (2 mm diameter) from paraffin blocks and arranged in new tissue microarray blocks.

RNA quality was assessed using an Agilent 2100 Bioanalyzer with an RNA 6000 Pico Chip (Agilent Technologies, Amstelveen, The Netherlands), and RNA quantification was performed using a NanoDrop 2000 Spectrophotometer system (Thermo Fisher Scientific, Waltham, MA, USA).

### 2.3. Library Preparation and Sequencing

Libraries were prepared for 50 bp single-end sequencing using a NEXTflex Small RNA-Seq Kit (PerkinElmer Inc., Waltham, MA, USA). Total RNA was extracted from FFPE samples using the ReliaPrep FFPE Total RNA MiniPrep Kit (Promega, Madison, WI, USA). Small RNAs were synthesized as single-stranded copy DNAs (cDNAs) using reverse transcription priming. The quality of these cDNA libraries was evaluated using a TapeStation 4200 (Agilent, Santa Clara, CA, USA), followed by quantification using the KAPA Library Quantification Kit (Kapa Biosystems, Wilmington, MA, USA) according to the manufacturer’s protocol. Sequencing was progressed as single-end (50 bp) using an Illumina NovaSeq6000 (Illumina, San Diego, CA, USA). The sequencing output was approximately 3 GB per sample.

### 2.4. Data Analysis

Low-quality bases or reads were trimmed or filtered using the Cutadapt tool [9]. The filtered reads were mapped to the reference genome of related species using the Bowtie aligner [10], followed by variant calling in the seed region of the miRNA. The mirdeep2 tool [11] was used to estimate the expression. Variant calling was performed using GATK [12] to search for variants in the miRNA seed regions. The miRNA expression level was measured with mirdeep2 [9] using the gene annotation database of the species, along with hairpin and mature miRNA sequence information, which was extracted from miBase [13]. Differentially expressed (DE) miRNAs were identified using the R package TCC [14], which applies robust normalization strategies to compare tag count data. DE miRNAs were identified based on a q-value threshold of less than 0.05 to correct errors caused by multiple testing [15].

In total, 10 websites containing information on miRNAs and targeted genes were searched to identify miRNA target genes. miRTarBase was used to construct the miRNA-target gene network (miRTarBase: a comprehensive database of miRNA target interactions (http://miRTarBase.cuhk.edu.cn/), accessed on 1 April 2023) [16].

By inputting the selected DE miRNAs, a miRNA-target gene network was constructed using Mirnet [17]. Subsequently, we performed functional annotation of the DE miRNAs using Gene Ontology (GO) [18] and pathway analysis using the Reactome [19] pathway database. We used the target genes identified from miR-net for gene-gene network analysis. Target genes were imported into Cytoscape [20], and STRING [21] was used to visualize interactions among these genes. The analysis was performed using medium confidence scores (0.4) and a significance threshold of *p* < 0.05. The pathways were enriched using nine databases, including GO and the Kyoto Encyclopedia of Genes and Genomes (KEGG) [22].

## 3. Results

### 3.1. Clinicopathological Data

The mean age of the patients in the SqCC metastasis and benign inflammatory groups was 63.6 years and 42 years, respectively. All patients in the SqCC metastasis group and 4 of 6 in the benign inflammatory necrosis group were male. The primary sites of SqCC included two tonsils, one tongue, one tongue base, one nasopharynx, and one unknown primary site. Five patients had tuberculosis, and one had Kikuchi-Fujimoto disease and inflammatory necrosis. The details are presented in Table 1. During the quality check after RNA extraction, one tumor sample from the nasopharynx failed; it was excluded from our study.

### 3.2. Identification of Differentially Expressed miRNAs

#### 3.2.1. Tumor and Tumor Necrosis (TN) versus Benign Inflammatory Necrosis (Benign) in Necrotic Lymphadenopathy

We divided the patients into three groups to analyze the expression of miRNAs: tumor, TN (necrosis of SqCC metastasis), and benign (benign inflammatory necrosis). We identified 3017 miRNAs. Principal component analysis was performed before analyzing differences in miRNA expression, and TN3 was excluded as an outlier. Finally, we analyzed the differences in miRNA expression between the five tumor groups, four TN groups, and six benign groups.

In total, 19 miRNAs were upregulated or downregulated in the tumor and TN groups compared to the benign group (fold change > 2, adjusted *p*-value < 0.05). Table 2 lists 17 upregulated (hsa-miR-31-5p, hsa-miR-141-3p, hsa-miR-149-5p, hsa-miR-182-5p, hsa-miR-183-3p, hsa-miR-183-5p, hsa-miR-200a-5p, hsa-miR-200a-3p, hsa-miR-200b-5p, hsa-miR-200b-3p, hsa-miR-200c-3p, hsa-miR-203a-3p, hsa-miR-205-5p, hsa-miR-429, hsa-miR-767-5p, hsa-miR-944, and hsa-miR-1-3p) and two downregulated miRNAs (hsa-miR-155-5p and hsa-miR-146b-5p).

Figure 2 shows the differential expression patterns of these miRNAs as a heatmap.

#### 3.2.2. Candidate Targets of miRNAs

In total, 3665 genes were commonly expressed in the miRWalk, miRanda, RNA22, and TargetScan databases, and 355 target genes were predicted to be targeted by two or more miRNAs. The STRING database was used to identify protein-protein interactions among these 355 targets. We defined proteins likely to interact with more than 10 other proteins as hub nodes. Figure 3 shows the protein-protein interactions derived from the 15 hub nodes (SMAD4, PARP1, FOXO1, CCND1, CREB1, CDKN1A, SMAD2, PTEN, AR, SP1, ETS1, RHOA, IGF1R, FOXO3, and IL6) in differentially expressed miRNAs between TN/tumor and benign tissues. The network between target genes was constructed using Cytoscape [20].

PTEN interacted with 26 other genes that were ranked first. The network between the candidate miRNAs and target genes is shown in Figure 4.

### 3.3. Cancer-Related GO Terms and KEGG Pathways Analysis

In the biologic process category of target genes that were differentially expressed between TN/tumor and benign, the top five GO terms were “regulation of cell proliferation”, “cell proliferation”, “negative regulation of cell proliferation”, “negative regulation of the apoptotic process”, and “negative regulation of programmed cell death” (Figure 5).

Table 3 summarizes the most enriched KEGG pathways in differentially expressed miRNAs between the TN/tumors and benign groups. Multiple cancer-related pathways such as “viral carcinogenesis”, “pathways in cancer”, “microRNAs in cancer”, and “p53 signaling pathway” were top-ranked in differently expressed miRNAs between the TN/tumor and benign groups.

## 4. Discussion

Central necrosis of metastatic lymph nodes is considered a poor prognostic indicator of head and neck squamous cell carcinoma (HNSCC) because it is related to massive cancer infiltration and intratumoral hypoxia-induced tumor necrosis [23,24]. Some studies have also shown an association between extracapsular spread and central necrosis [25,26]. Therefore, the exact diagnosis of necrotic lymph nodes is important. miRNA is one of the well-studied biomarkers extensively researched for their potential to enhance diagnostic accuracy. miRNA possesses several merits as a biomarker. It is synthesized quickly in response to a pathological situation, is highly specific, and remains in the system for an extended period, making it easily detectable due to its presence in the plasma [27]. For those reasons, various studies on the clinical potential of miRNAs as diagnostic biomarkers have been performed for various types of cancers with respect to FFPE samples and circulating miRNAs in saliva and blood.

Necrosis can be described as a pathological process of cell death. It also causes inflammation of the surrounding tissues. Therefore, small biopsies such as FNAC or CNB can only present inflammatory cells or limited cellularity. A previous study reviewed cases that revealed necrotic features in FNAC from cervical lymph nodes [28]. They reviewed 460 patients, and tuberculosis was the most common etiology in the final diagnosis. Notably, 49 of 460 cases were finally confirmed as malignancy. However, the FNAC of tuberculosis lymphadenitis showed a relatively high specificity of 88–96% [29,30]. When using PCR, Xpert MTB/Rif, and Xpert, the sensitivity and specificity were reported to be 78% and 90–100%, and 78% and 38–87%, respectively [31]. Small biopsies can easily detect many cases of tuberculosis lymphadenitis, but in severe necrosis cases, the diagnosis cannot be completed.

In our study, miRNAs showed good prediction for SqCC even in necrotic tissues, differing from benign inflammatory necrosis. The role of miRNAs was usually defined as oncogenes and suppressor genes in HNSCC. A recent study reviewed upregulated and downregulated miRNAs in HNSCC [32] and showed multiple well-studied miRNAs in HNSCC. Among them, hsa-miR-1, hsa-miR-31, hsa-miR-141, hsa-miR-146, hsa-miR-155, hsa-miR-200, hsa-miR-203, and hsa-miR-205 were commonly downregulated in the tumor necrosis group compared to those in the benign inflammatory necrosis group. The miR-200 family includes miR-200a, miR-200b, miR-200c, miR-141, and miR-429 [33], all of which play important roles as tumor suppressors by repressing malignant cell transformation and inhibiting tumor initiation [34,35]. In HNSCC, the miR-200 family has been reported to negatively modulate the progress of epithelial-mesenchymal transition [36]. miR-31 is highly expressed, especially in SqCC, and its expression is a potential marker of metastasis or poor prognosis [37]. Previous studies have revealed that the expression levels of miR-31 are increased in various HNSCC samples (saliva, plasma, and tumor tissues) and are positively correlated with poor pathological parameters [38,39] and advanced staging [40]. miR-203 and miR-205 are well-known tumor suppressors in HNSCC. In some studies on laryngeal cancer, miR-203 and miR-205 were shown to play a role in inhibiting invasion and inducing apoptosis [41,42].

Notably, the expression of miR-21, the most extensively studied oncomiR in HNSCC, did not differ between TN and benign tumors. miR-21 has been proposed as a biomarker of diagnostic, prognostic, and therapeutic value in HNSCC. However, the expression of miR-21 was high in both the TN and benign groups. Most cases in our benign inflammatory necrosis group were tuberculosis-induced necrosis, and miR-21 plays an important role in tuberculosis by inhibiting apoptosis [43]. Similarly, miR-155 is known to be upregulated in HNSCC. However, in our study, its expression was slightly downregulated compared to the benign group. miR-155 expression is also upregulated in tuberculosis by inhibiting apoptosis [44] and enhancing autophagy [45].

A study that examined the expression of miRNAs in SqCC metastatic FFPE samples and FNAC of patients with HNSCC [46] found seven highly upregulated miRNAs in metastatic lymph node samples from patients with HNSCC. Among these, miR-200a, miR-200c, miR-203, and miR-205 were highly upregulated in our samples, and miR-203 and miR-205 were validated in FNAC samples. In that study, it was considered that even if aspirated samples did not contain metastatic cells, FFPE samples harboring micro-metastases could be correctly identified using molecular markers. These results are consistent with ours. Therefore, our necrotic samples might harbor micro-metastases that can be detected using molecular markers. When necrosis was severe and there was absolute acellular status, miRNA expression could not be checked. In our study, although we selected only the necrotic part of the samples using microscopes, RNA extraction was performed, and there might have been remnant micro-metastatic information. In clinical settings, the tumor could be diagnosed if the aspirated or core needle biopsied sample had a small amount of viable cells along the necrotic tissues.

Our study had some limitations. First, the sample size was small. It was difficult to select samples where only the necrotic portion that was not contaminated by cancer cells could be resected. Furthermore, the necrotic portions have scant RNAs, so some cases were excluded due to RNA quality control. Many cases of benign inflammatory necrosis did not need resection because most of them were treated and diagnosed with only small biopsies and medications. Therefore, resection samples were only obtained when the small biopsy diagnosis was difficult or the case did not respond to anti-tuberculosis medications. Second, we did not verify steps that can confirm the effectiveness of our target miRNAs in other necrotic tissues.

Nevertheless, this study has several advantages. There have been few studies on miRNA profiling of the necrotic portion during tumor necrosis. In addition, we could identify information about tumor micro-metastases only from necrotic samples, which could help develop new target miRNAs to increase the diagnostic efficacy of FNAC.

## 5. Conclusions

Our study revealed significant differences in miRNA expression between benign inflammation and TN despite having the same necrotic histopathology. We identified 19 miRNAs that exhibited significant differential expression between benign and necrotic tumors, many of which are known to be associated with head and neck SqCC miRNAs. These target miRNAs would potentially aid in the differential diagnosis of necrotic lymph nodes when severe necrosis makes it difficult to determine the underlying pathology. Further large-scale or extended studies are required to validate our findings and support their clinical utility.

## Figures and Tables

**Figure 1 biomedicines-11-02407-f001:**
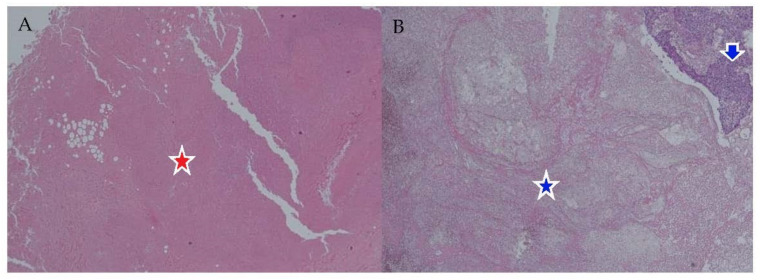
Histopathologic slides of necrotic lymphadenopathy stained by H&E (×40): Necrosis in tuberculosis [red star] (**A**) and necrotic portion induced by metastatic lymphadenopathy (necrotic portion, blue star; metastasis, blue arrow) (**B**).

**Figure 2 biomedicines-11-02407-f002:**
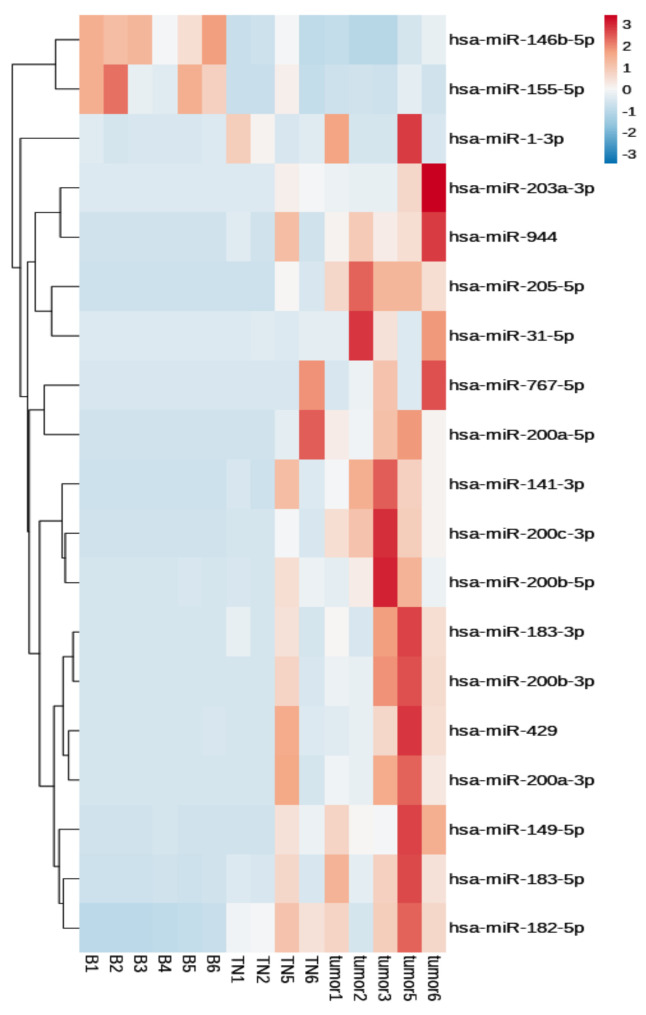
Heatmap of target microRNAs that showed differences in expression between tumor necrosis and benign inflammatory necrosis. TN, tumor necrosis; B, benign inflammatory necrosis; miR, microRNA.

**Figure 3 biomedicines-11-02407-f003:**
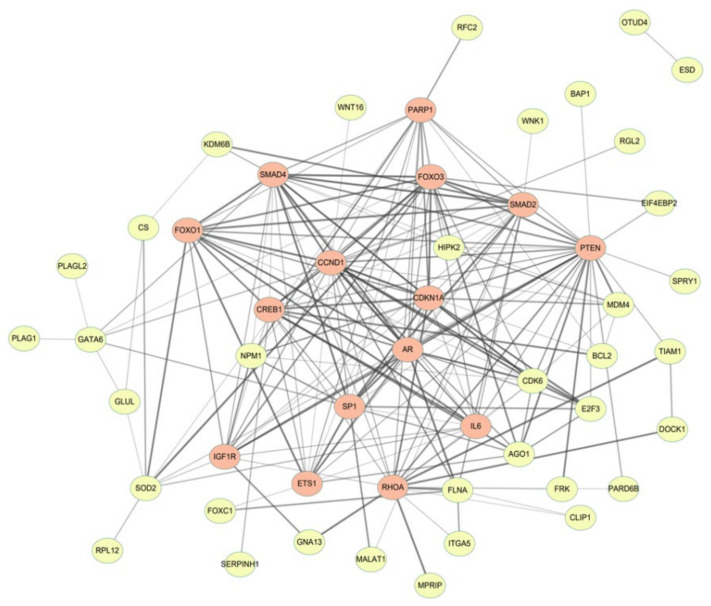
The interaction network derived from the 15 hub nodes for the targets of the differentially expressed miRNAs between the tumor necrosis portion and benign inflammatory necrosis portion was presented from the STRING database (https://string-db.org/) on Cytoscape software (version 3.8.2, https://cytoscape.org/).

**Figure 4 biomedicines-11-02407-f004:**
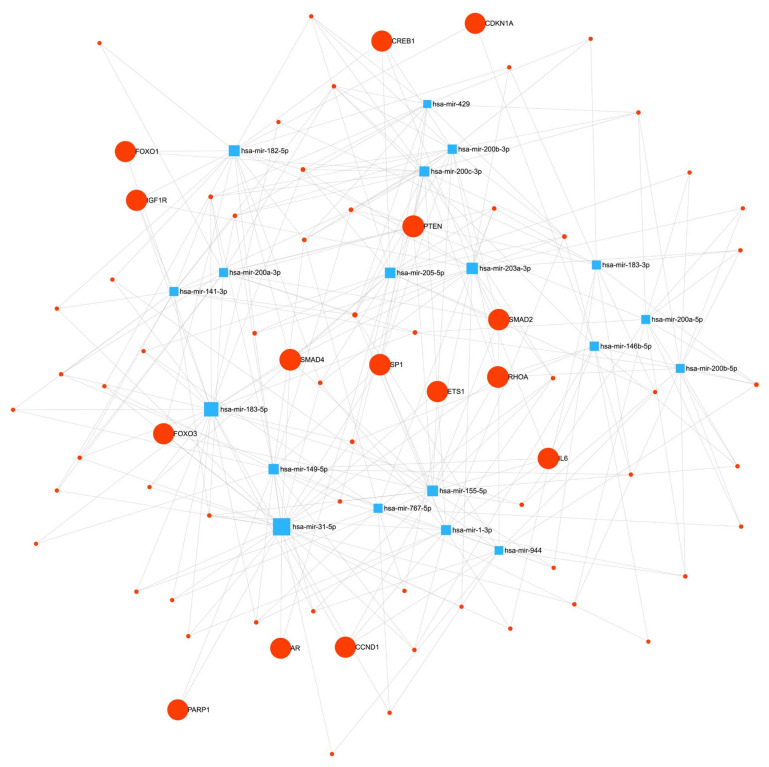
The miRNA-target gene network derived from the differentially expressed miRNAs from the tumor necrosis portion and benign inflammatory necrosis portion from the miRNet software (version 2.0, https://www.mirnet.ca/).

**Figure 5 biomedicines-11-02407-f005:**
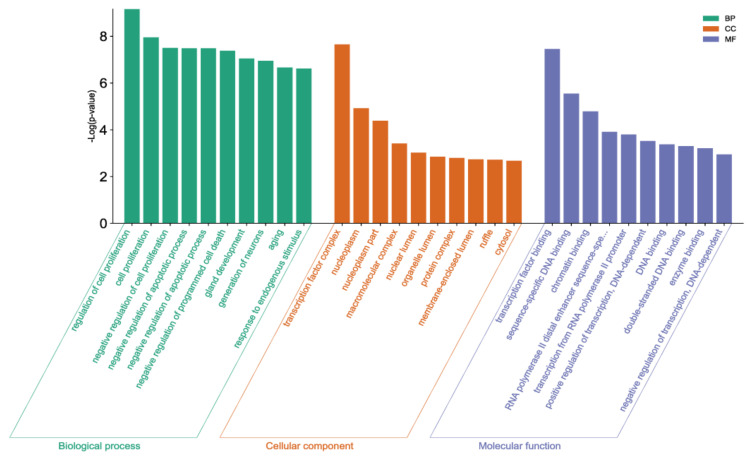
Gene Ontology enrichment of miRNAs that were differentially expressed in two groups: tumor necrosis portion and benign inflammatory necrosis portion.

**Table 1 biomedicines-11-02407-t001:** Clinicopathologic characteristics of the study participants.

	SqCC Metastasis	Benign Inflammatory Necrosis
Mean age (range)	63.6 (52–73)	42 (24–55)
Sex (male:female)	6:0	4:2
Size (cm, range)	3.24 (2–5.3)	2.75 (2.3–3.3)
Primary site of SqCC		NA
Tongue	1	
Tongue base	1	
Tonsil	2	
Nasopharynx	1	
Cancer of Unknown Primary (CUP)	1	
Pathology	NA	
Tuberculosis		5
Kikuchi-Fujimoto disease		1
Total	6	6

SqCC, squamous cell carcinoma; NA, not applicable.

**Table 2 biomedicines-11-02407-t002:** Significantly differentially expressed miRNAs between tumor and tumor necrosis (TN) versus benign inflammatory necrosis.

miRNAs, Tumor, TN vs. Benign	Fold Change	Adjusted *p*-Value
Upregulated		
hsa-miR-31-5p	3.95	0.054
hsa-miR-141-3p	8.57	<0.001
hsa-miR-149-5p	5.00	0.024
hsa-miR-182-5p	4.33	0.004
hsa-miR-183-3p	6.57	0.019
hsa-miR-183-5p	3.74	0.008
hsa-miR-200a-3p	6.52	>0.05
hsa-miR-200a-5p	8.94	>0.05
hsa-miR-200b-3p	5.64	0.002
hsa-miR-200b-5p	5.88	0.007
hsa-miR-200c-3p	4.68	0.002
hsa-miR-203a-3p	8.87	<0.001
hsa-miR-205-5p	7.68	<0.001
hsa-miR-429	6.29	0.015
hsa-miR-767-5p	7.76	0.010
hsa-miR-944	6.51	0.021
hsa-miR-1-3p	2.90	0.034
Downregulated		
hsa-miR-155-5p	−2.35	0.048
hsa-miR-146b-5p	−1.91	0.073

TN, tumor necrosis; Benign, benign inflammatory necrosis.

**Table 3 biomedicines-11-02407-t003:** KEGG pathway analysis of the differentially expressed miRNAs and targets in two groups.

KEGG Pathway	*p*-Value	Numbers of Involved Genes	Numbers of Involved miRNAs
Adherens junction	<1.00 × 10^−325^	40	9
Viral carcinogenesis	1.11 × 10^−15^	84	8
Pathways in cancer	2.41 × 10^−12^	136	8
MicroRNAs in cancer	5.95 × 10^−11^	57	4
Oocyte meiosis	1.68 × 10^−8^	50	7
Hippo signaling pathway	2.23 × 10^−8^	49	6
Hepatitis B	6.80 × 10^−7^	64	6
Bacterial invasion of epithelial cells	1.14 × 10^−6^	34	6
p53 signaling pathway	2.07 × 10^−6^	33	5
Ubiquitin mediated proteolysis	3.16 × 10^−6^	52	4
Prostate cancer	4.84 × 10^−6^	49	6
FoxO signaling pathway	5.27 × 10^−6^	57	5
Chronic myeloid leukemia	1.37 × 10^−5^	38	7
Lysine degradation	1.80 × 10^−5^	14	7
Cell cycle	6.05 × 10^−5^	47	3
Endocytosis	0.000014782	52	3
Colorectal cancer	0.000913152	28	4
Protein processing in the endoplasmic reticulum	0.001171519	52	4

## Data Availability

Not applicable.

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
