# Peer review of "Does the Necrotic Portion of Metastatic Lymphadenopathy from Squamous Cell Carcinoma Still Have Tumoral Oncologic Information? Differential Diagnosis of Benign Necrotic Lymphadenopathy Using microRNA"

_biomedicines, 2023, doi:10.3390/biomedicines11092407_

Round 1

Reviewer 1 Report

In this study, the authors aimed to explore the possible use of the necrotic portion of metastatic lymphadenopathy in the diagnosis of squamous cell carcinoma by profiling miRNAs. Although this study has interests and their experiments and analyses have been properly conducted, the major issue is that there are no clear conclusions to answer or respond their research goals presented in the title. In addition to this major issue, several concerns are also need to be addressed.

1. In the recruited samples, samples with SqCC metastasis collected during 2014-2022 were all from males. Why SqCC metastasis samples from females are rare? In addition, are expression profiles of these miRNAs similar between males and females? Authors may further address the rationale.

2. In Table 1, the mean age of samples between SqCC metastasis and benign inflammatory necrosis group appears significantly different. Is it difficult to acquire samples with similar mean ages? If it is, authors may further address

3. In Fig. 1A, necrosis in the tuberculosis is suggested to be indicated with an arrow, the red star may cover the lesion area and appear confused.

4. Based on the changes of miRNAs and genes, is there a specific panel of miRNAs that can be used to diagnose or predict SqCC from lymphadenopathy samples?

5. There are still typos (e.g. line 235, “MiRNA”; ) and incorrect formats (e.g. line 141-146, the text should be regular rather than italic).

Minor editing of English language required.

Author Response

In this study, the authors aimed to explore the possible use of the necrotic portion of metastatic lymphadenopathy in the diagnosis of squamous cell carcinoma by profiling miRNAs. Although this study has interests and their experiments and analyses have been properly conducted, the major issue is that there are no clear conclusions to answer or respond their research goals presented in the title. In addition to this major issue, several concerns are also need to be addressed.

- Response: I agreed that out study has some limitations. We cannot conclude the clear comments because of small number of specimen. In our study, we find candidate of target microRNAs for identifying SqCC metastasis from necrotic changes, and further study is needed to validate the target microRNAs.

#1. In the recruited samples, samples with SqCC metastasis collected during 2014-2022 were all from males. Why SqCC metastasis samples from females are rare? In addition, are expression profiles of these miRNAs similar between males and females? Authors may further address the rationale.

- Response: Thank you for your insightful questions. There was no other reason that all patients with SqCC were males. As you know, the incidence of head and neck squamous cell carcinoma in males is higher than in females. In a recent study that analyzed the prevalence of head and neck cancers in 10 million healthy people during a 10-year period, the male-to-female ratio of cancer incidence was 1.0 among people in their 20s, 1.5 among those in their 30s, 2.5 in the 40s, 4.4 in the 50s, 5.4 in the 60s, and 4.6 in the 70s [1]. Our study limited the inclusion criteria: severe necrosis where dissection without contaminating the cancer cells was possible. Therefore, our sample included only males.
As you pointed out, several miRNAs might be sex-biased. Previous studies of miRNAs in head and neck cancer also included a predominantly male study population (79.2% males [2], 97.5% males [3]. We believe the sex differences would not influence the results of our study.

  1. In Table 1, the mean age of samples between SqCC metastasis and benign inflammatory necrosis group appears significantly different. Is it difficult to acquire samples with similar mean ages? If it is, authors may further address

- Response: Thank you for your kind review. We agree that it would be better for the groups to have similar general characteristics such as sex and age. As you know, the incidence of head and neck cancer is high in patients aged over 60 [1]. Further, tuberculosis lymphadenitis and Kikuchi lymphadenitis showed high incidence in relatively younger patients.
In our treatment protocol, we did not excise benign inflammatory necrosis as a first treatment. We performed fine needle aspiration cytology for tuberculosis lymphadenitis. If it got diagnosed, then we treated it with anti-tuberculosis medicines. Therefore, a few cases of resected tuberculosis specimens were not diagnosed using FNA or not treated with anti-tuberculosis medicine. We only included samples with severe necrosis. 

  1. In Fig. 1A, necrosis in the tuberculosis is suggested to be indicated with an arrow, the red star may cover the lesion area and appear confused.

- Response: We agree that using the same color for the star can be confusing. We have changed the color of the necrotic portion of SqCC.

  1. Based on the changes of miRNAs and genes, is there a specific panel of miRNAs that can be used to diagnose or predict SqCC from lymphadenopathy samples?

- Response: In our study, we identified 19 miRNAs. Compared to a previous study [2], the potent diagnostic panel can be miR-200a, miR-200c, miR-203, and miR-205. However, further study and validation studies are needed. 

  1. There are still typos (e.g. line 235, “MiRNA”; ) and incorrect formats (e.g. line 141-146, the text should be regular rather than italic).

-  Response: Thank you for the kind review. We have corrected the errors, as you pointed out.

Reference

  1. Park, J.O.; Nam, I.C.; Kim, C.S.; Park, S.J.; Lee, D.H.; Kim, H.B.; Han, K.D.; Joo, Y.H. Sex differences in the prevalence of head and neck cancers: a 10-year follow-up study of 10 million healthy people. Cancers (Basel). 2022, 14(10), 2521.
  2. de Carvalho, A.C.; Scapulatempo-Neto, C.; Maia, D.C.; Evangelista, A.F.; Morini, M.A.; Carvalho, A.L.; Vettore, A.L. Accuracy of microRNAs as markers for the detection of neck lymph node metastases in patients with head and neck squamous cell carcinoma. BMC Med. 2015, 13, 108.
  3. Chang, Y.A.; Weng, S.L.; Yang, S.F.; Chou, C.H.; Huang, W.C.; Tu, S.J.; Chang, T.H.; Huang, C.N.; Jong, Y.J.; Huang, H.D. A three–microRNA signature as a potential biomarker for the early detection of oral cancer. Int J Mol Sci.201819, 758. 

Reviewer 2 Report

This manuscript describes bioinformatics analysis of miRNAs derived from necrotic lymph nodes of patients with either SCC or benign inflammatory conditions. The study itself includes a very small number of cases but appears to have been performed rigorously, and results have been clearly presented. There are shortcomings of the work, principally that none of the results have been tested or validated, which would strengthen the article, given that miRNA profiling is notoriously unreproducible and the manuscript is based on very few cases. As a means of hypothesis generation, the methodology is appropriate. However, at this stage, and due to the highly selected nature of the samples used (“pure” necrosis), it is difficult to see future applications of the findings. Overall, the manuscript is very well written. English language is excellent, and both figures and tables are very well constructed and appropriate for this type of research. Referencing could be improved (see below), but this is easily fixed.

1. The authors have extensively discussed the roles of miRNAs in terms of viable and actively growing malignancy as well as malignant processes, despite the source material of the miRNAs being necrotic, largely acellular tissue. Although miRNAs are considered to be long-lived in the circulation, have the authors considered that the miRNAs in the necrotic lymph nodes may reflect the process of necrosis? Most, if not all bioinformatics and analysis programmes are built on the functions of live cells, and choosing results that fit with those programmes may not represent the tissues that they are analysing in this study. Could the authors reconsider the analysis and interpretation of their data in terms of the (necrotic) tissue from which they derived the source material (miRNAs) rather than what the tissue might have been at some stage in the past?

2. This study has used very small numbers of samples and these samples are highly selected according to very strict criteria regarding tissue organisation and pathological features. Because the study is based on profiling rather than the presence or absence of a miRNA or a mutation (for example), how do the authors envisage their findings to be translatable to clinical practice where tissues are more heterogeneous and not as clearly demarcated?

3. Because this study is limited to hypothesis generation on a very small number of specimens, I feel that conclusions such as the last sentence in the Abstract (and elsewhere) overstate the findings and/or their potential future use and could be modified to better reflect the content of the manuscript.

4. Referencing: This manuscript is very well written and was a pleasure to read, however it is quite noticeable that references have not been cited in relation to a number of major statements. For example, in the final paragraph of the Introduction, the investigators describe commercial use of miRNAs but include no references. Could the authors please go through their manuscript and add the relevant references?

5. Lines 143-144: Did the authors want to add in the single nasopharynx primary site of SqCC to complete their list of the 6 cases?

6. Section 3.1: This section has accidentally been written in italics rather than normal script.

7. Line 141: metastasis and benign inflammatory groups (‘and’ is missing)

8. Line 194: which were differentially expressed (‘were’ is missing)

Author Response

- We thank the reviewer for the kind and detailed review. We have implemented the recommendations accordingly. The comments helped make our manuscript more comprehensive and valuable.

This manuscript describes bioinformatics analysis of miRNAs derived from necrotic lymph nodes of patients with either SCC or benign inflammatory conditions. The study itself includes a very small number of cases but appears to have been performed rigorously, and results have been clearly presented. There are shortcomings of the work, principally that none of the results have been tested or validated, which would strengthen the article, given that miRNA profiling is notoriously unreproducible and the manuscript is based on very few cases. As a means of hypothesis generation, the methodology is appropriate. However, at this stage, and due to the highly selected nature of the samples used (“pure” necrosis), it is difficult to see future applications of the findings. Overall, the manuscript is very well written. English language is excellent, and both figures and tables are very well constructed and appropriate for this type of research. Referencing could be improved (see below), but this is easily fixed.

  1. The authors have extensively discussed the roles of miRNAs in terms of viable and actively growing malignancy as well as malignant processes, despite the source material of the miRNAs being necrotic, largely acellular tissue. Although miRNAs are considered to be long-lived in the circulation, have the authors considered that the miRNAs in the necrotic lymph nodes may reflect the process of necrosis? Most, if not all bioinformatics and analysis programmes are built on the functions of live cells, and choosing results that fit with those programmes may not represent the tissues that they are analysing in this study. Could the authors reconsider the analysis and interpretation of their data in terms of the (necrotic) tissue from which they derived the source material (miRNAs) rather than what the tissue might have been at some stage in the past?

- Response: As you mentioned, necrosis is a pathological cell death process. Massive infiltration of cancer caused intratumoral hypoxia and central necrosis of metastatic lymph nodes. Therefore, we thought necrosis of metastatic lymph nodes would have micrometastases which cannot be identified using a microscope.
MicroRNAs can be affected by many circumferential effects. We agree that necrosis also affects their expressions. Therefore, we compared the same conditions of necrotic tissues with different causes of cancer and benign inflammation.

  1. This study has used very small numbers of samples and these samples are highly selected according to very strict criteria regarding tissue organisation and pathological features. Because the study is based on profiling rather than the presence or absence of a miRNA or a mutation (for example), how do the authors envisage their findings to be translatable to clinical practice where tissues are more heterogeneous and not as clearly demarcated?

- Response: We apologize for the small number of study samples. We hypothesized that the conditions were not identified using a microscope. However, microRNAs can be checked for differential diagnosis. In clinical findings showing contaminated cancer cells, the yield of microRNAs can be increased, and we can get more exact results. Further studies are needed to validate our target microRNAs.

  1. Because this study is limited to hypothesis generation on a very small number of specimens, I feel that conclusions such as the last sentence in the Abstract (and elsewhere) overstate the findings and/or their potential future use and could be modified to better reflect the content of the manuscript.

- Response: We agree with your opinion. We suggested the possibility of using target microRNAs. As you mentioned, we have corrected our statements.

  1. Referencing: This manuscript is very well written and was a pleasure to read, however it is quite noticeable that references have not been cited in relation to a number of major statements. For example, in the final paragraph of the Introduction, the investigators describe commercial use of miRNAs but include no references. Could the authors please go through their manuscript and add the relevant references?

- Response: Thank you for pointing this out. We have reviewed the text and added the relevant references accordingly.

  1. Lines 143-144: Did the authors want to add in the single nasopharynx primary site of SqCC to complete their list of the 6 cases?

- Response: Thank you for your kind review. We have corrected this error.

  1. Section 3.1: This section has accidentally been written in italics rather than normal script.

- Response: Thank you for your kind review. We have corrected this error.

  1. Line 141: metastasis and benign inflammatory groups (‘and’ is missing)

- Response: Thank you for your kind review. We have corrected this error.

  1. Line 194: which were differentially expressed (‘were’ is missing)

- Response: Thank you for your kind review. We have corrected this error.

Round 2

Reviewer 1 Report

The previous issues have been addressed and the manuscript is also improved. Despite the limited results of this study, this study may provide some seminal contributions to exploring the potential of miRNAs as diagnostic markers for squamous cell carcinoma.

Reviewer 2 Report

The authors have addressed reviewers’ comments and I feel that the manuscript is suitable for publication.

1. Following principal component analysis, the authors excluded one of the 5 tumour necrosis samples from their analysis (TN3, section 3.2.1). Out of interest, if the authors applied the results that they derived in subsequent analyses to this sample, would they have categorised it correctly?

2. In a clinical sense, do the authors feel that necrosis in a lymph node would result from either a benign or a malignant underlying pathology (but not both)? For example, in areas where tuberculosis is endemic, do the authors expect that necrosis in the lymph node of a patient with both tuberculosis and SCC would be due to either the underlying tuberculosis or to the underlying SCC?

3. Line 327-328: ‘along with the necrotic tissues’ (‘with’ is missing).

4. Lines 346-347: should this phrase be ‘benign and tumor-associated necrosis’?